# Dopamine D1 Receptor in Cancer

**DOI:** 10.3390/cancers12113232

**Published:** 2020-11-02

**Authors:** Paweł Sobczuk, Michał Łomiak, Agnieszka Cudnoch-Jędrzejewska

**Affiliations:** 1Department of Experimental and Clinical Physiology, Laboratory of Centre for Preclinical Research, Medical University of Warsaw, 02-097 Warsaw, Poland; lomiak.m@gmail.com (M.Ł.); agnieszka.cudnoch-jedrzejewska@wum.edu.pl (A.C.-J.); 2Department of Soft Tissue/Bone Sarcoma and Melanoma, Maria Sklodowska-Curie National Research Institute of Oncology, 02-097 Warsaw, Poland

**Keywords:** dopamine, D1 receptor, cancer, target, pathways, signaling

## Abstract

**Simple Summary:**

Circulating hormones and their specific receptors play a significant role in the development and progression of various cancers. This review aimed to summarize current knowledge about the dopamine D1 receptor’s biological role in different cancers, including breast cancer, central nervous system tumors, lymphoproliferative disorders, and other neoplasms. Treatment with dopamine D1 receptor agonists was proven to exert a major anti-cancer effect in many preclinical models. We highlight this receptor’s potential as a target for the adjunct therapy of tumors and discuss possibilities and necessities for further research in this area.

**Abstract:**

Dopamine is a biologically active compound belonging to catecholamines. It plays its roles in the human body, acting both as a circulating hormone and neurotransmitter. It acts through G-protein-coupled receptors divided into two subgroups: D1-like receptors (D1R and D5R) and D2-like receptors (D2R, D3R, D4R). Physiologically, dopamine receptors are involved in central nervous system functions: motivation or cognition, and peripheral actions such as blood pressure and immune response modulation. Increasing evidence indicates that the dopamine D1 receptor may play a significant role in developing different human neoplasms. This receptor’s value was presented in the context of regulating various signaling pathways important in tumor development, including neoplastic cell proliferation, apoptosis, autophagy, migration, invasiveness, or the enrichment of cancer stem cells population. Recent studies proved that its activation by selective or non-selective agonists is associated with significant tumor growth suppression, metastases prevention, and tumor microvasculature maturation. It may also exert a synergistic anti-cancer effect when combined with tyrosine kinase inhibitors or temozolomide. This review provides a comprehensive insight into the heterogeneity of dopamine D1 receptor molecular roles and signaling pathways in human neoplasm development and discusses possible perspectives of its therapeutic targeting as an adjunct anti-cancer strategy of treatment. We highlight the priorities for further directions in this research area.

## 1. Introduction

Dopamine (DA) exerts its major functions in organisms, acting both as a neurotransmitter in the central nervous system and as a circulating hormone in the periphery. Peripheral synthesis of this compound can occur in adrenal glands, spleen, pancreas, and sympathetic nerves [1,2]. Central actions of DA include the regulation of emotions [3], motivation [3], addiction [4], movement [5] and cognition [6], while peripherally influencing kidney functions [7], blood pressure regulatory mechanisms [7], and even immunological response in humans [8]. The expression of dopamine receptors (DRs) was described in a wide range of tissues and organs such as the brain, retina, heart, coronary arteries, gastrointestinal tract, and sympathetic ganglia [9]. The dysregulation of the dopaminergic system is a fundamental mechanism of many diseases, including Parkinson’s disease, schizophrenia [9], hypertension and metabolic dysfunctions [10] or gut motility abnormalities [11]. Dopamine receptors have been found as targets for the treatment of those pathologies [12,13].

Epidemiological studies showed that cancer incidence among patients suffering from diseases associated with dopamine system dysregulation could be different comparing to a healthy population. The population of schizophrenic patients has a lower incidence of colorectal or prostate cancer, but also opposite conclusions were made, for instance, regarding breast cancer [14,15]. Parkinson’s disease patients were shown to have a declined risk of developing various cancers, especially smoking-related neoplasms, such as lung or bladder cancer [16,17]. On the contrary, those patients are more inclined toward developing skin tumors (including melanoma) and breast cancer [18]. DRD1 gene polymorphism (G allele rs686) is a predisposing factor for developing non-small-cell lung cancer among passive smokers during childhood [19]. All those observations might suggest a possible connection between the dopaminergic system and carcinogenesis.

Furthermore, many in vivo and in vitro studies have discussed the significance of dopamine in proliferation, apoptosis, tumor angiogenesis, and drug resistance among different cancers, including gliomas [20], gastric cancer [21], breast cancer [22] or ovarian cancer [23]. Animal models with modified dopaminergic systems showed this hormonal system’s modulatory role in tumor growth [24,25]. From the perspective of rapidly widening literature regarding the value of dopamine receptors in cancer pathogenesis, we reviewed the current knowledge about the activity and signaling mechanisms of D1 receptor (D1R) in different cancers.

## 2. D1 Receptor and Its Signaling Pathways

The human dopamine D1 receptor consists of 446 amino acids. It is a member of the heptahelical G-protein coupled receptor (GPCR) superfamily, divided into two groups: D1-like receptors (D1 and D5 receptors) and D2-like receptors (D2, D3, D4 receptors) [26,27]. GPCRs have a trans-membrane α-helical configuration consisting of seven membrane-spanning helices, three extracellular loops, and three intracellular loops [28]. Even though this class of receptors’ proposed division is widely acknowledged, the existence of heteromeric dopamine receptors complexes needs to be considered because of the unique pharmacological properties of such dimers [29,30]. Discussed D1 receptors can be located only in postsynaptic dopamine-mediated cells [31]. They are encoded by the DRD1 gene that is located on 5q35.1 chromosome [32,33].

Signaling pathways of D1R are mainly based on G protein activity, but they also contain G protein-independent downstream signaling (Figure 1). It is commonly described that the activation of D1R coupled to G_s_α protein leads to the increased synthesis of the secondary messenger—cAMP—which stimulates the activity of phosphokinase A (PKA) [27,33,34]. PKA can interact with several targets, including cAMP response element-binding protein (CREB), glutamate receptors, GABA receptors, or ion channels. Moreover, cAMP-regulated phosphoprotein 32-kDa (DARPP-32) is also a molecule targeted and phosphorylated by PKA. The phosphorylated form of DARPP-32 (Thr34) shows inhibitory effects on protein phosphatase 1 (PP1), a histone modulator enhancing the efficacy of the PKA pathway [35,36]. On the contrary, phosphorylation at Thr-75 transforms DARPP-32 into an inhibitor of PKA [37]. Interestingly, DARPP-32 overexpression has a pro-cancer activity in colorectal cancer [38], non-small cell lung carcinoma [39], gastric cancer [40] and has even been discussed as a potential anti-cancer target [41]. Contrarily, the absence of DARPP-32 expression has been associated with worse prognosis in esophageal squamous cell carcinoma and oral premalignant and malignant lesions [42,43,44].

Another pathway engaged in response to D1R activation is associated with the G_q_α protein and activation of phospholipase C (PLC), which can subsequently catalyze the production of inositol triphosphate (IP3) and diacylglycerol (DAG). These two increase intracellular calcium levels and activate protein kinase C (PKC), respectively (Figure 1) [45,46].

One of the most important actions of D1 receptors in the context of tumor development is the involvement in long-term events (Figure 1). Among them is the MAPK (mitogen-activated protein kinase) pathway. This receptor’s crucial role in MAPK-associated signaling was shown in the modulation of the extracellular-signal-regulated kinases (ERK), especially ERK1/2, resulting in various effects like cell proliferation, apoptosis, and differentiation [47,48]. Distinct, not involving G protein activation, the pathway triggered upon the activation of D1R in certain conditions is the Akt-GSK3 pathway. The Akt-GSK3 regulates cell survival, maturation, and the transcription of various genes, including those encoding cyclin D_1_, p53, and c-Myc proteins [49,50].

Novel insight into downstream D1 signaling pathway components, such as G-protein-independent β-arrestins and G-protein-coupled receptor kinases, allowed to distinguish so-called non-canonical pathways. One of them is cGMP signaling, a potential therapeutic target widely discussed in breast cancer [51]. D1R activation in this type of cancer was proven to trigger soluble guanylate cyclase (sGC) production that can convert GTP into cGMP. Consequently, cGMP as a second messenger activates phosphokinase G (PKG) that plays a significant role in the effects of D1 receptor activation on cell viability, proliferation, and apoptosis [52].

## 3. Effects of D1 Receptor in Various Tumor Types

Gradually increasing evidence regarding the value and potential clinical utility of D1R leads to distinguishing this receptor’s cancer-specific actions. Alterations in this receptor’s expression have been described in various neoplasms (Table 1 and Appendix A). Heterogeneity of both roles and mechanistic background of D1R modulation requires an appropriate approach, given different tumors’ biology. In this review, we decided to summarize and discuss D1R significance by comprehensively presenting the current knowledge divided depending on the cancer type regarding breast cancer, nervous system neoplasms, lymphoproliferative disorders, gastrointestinal tract tumors, and findings in the other cancer types.

### 3.1. Breast Cancer

#### 3.1.1. In Vitro Studies

Firstly, Johnson et al. [54] showed that D1 receptor agonist SKF38393 reduced the viability of growth-inducted MCF7 ER-positive breast cancer cell line stronger than tamoxifen (selective estrogen receptor modulator used in the treatment of breast cancer). The phenomenon of dopamine agonist’s effect on estradiol-stimulated ER-positive cell lines is environment-dependent and occurs only upon the hormonal induction of growth [52]. This observation may result from ER’s inactivation via a cross-talk mechanism, in this case, mediated by D1 receptor referred to as antiestrogen binding sites (AEBS) hypothesis [63,64].

Treatment with D1R agonists (SKF38393, A68930, A77636, fenoldopam) was shown to dramatically suppress the viability of triple-negative breast cancer cell lines (MDA-MB-231, MDA-MB-468, SUM149) and hormonal breast cancer cell line (BT-20) without affecting the proliferation of normal breast epithelium [52,55,65]. D1R activation increases the cGMP level in triple-negative breast cancer and leads to a cAMP level decrease even though D1R is traditionally known as a cAMP/PKA pathway activator [48,66,67]. The observed decrease in cAMP concentration is probably due to phosphodiesterases’ (PDEs’) activation secondary to cGMP elevation, indicating a reciprocal relationship between these cyclic nucleotides [68]. Finally, levels of phosphorylated ERK1/2, Akt, and CREB were elevated after treatment with D1R agonist. The modulation of these compounds by the D1 receptor agonist was previously described in tumorous and non-tumorous cells, including neuroblastoma, astrocytes, or macrophages [69,70,71]. D1R activation in triple-negative breast cancer cell line (4T1.2) leads to the induction of apoptosis (increased level of cleaved caspase 3), autophagy (elevation of LC3-phosphatidylethanolamine conjugate–LC3A/B II), and the phosphorylation of eukaryotic translation initiation factor 2-alpha (eIF2a) [55,72,73]. The increase in phospho-eIF2a level may contribute to caspase cleavage and subsequent apoptosis [74].

Selective D1 receptor agonist fenoldopam and l-SPD (l-stepholidine), D1R agonist and D2R antagonist, display inhibitory potential on metastatic breast cancer cell lines MDA-MB-231 and 4T1 [53,75]. Notably, both agents were used in very high concentrations (0.39–100 µM). Compared with contrary data from Borcherding et al. [52] where fenoldopam strongly suppressed cell in nanomolar concentrations, we can conclude that the D1R-mediated induction of apoptosis is most effective upon treatment longer than 48 h.

Targeting D1R was shown to enhance sunitinib efficacy, which implies the possible utility of D1 agonists to partially overcome breast cancer resistance to already known chemotherapeutics partially. Combined therapy may decrease the tyrosine kinase inhibitors (TKIs) dosage, reduce its side effects (e.g., cardiotoxicity or skin toxicity) [76], and lower the risk of acquired resistance. Wang et al. observed the synergistic effect of dopamine and sunitinib, a multi-targeted receptor TKI, on drug-resistant MCF-7/Adr cell line. It resulted in a significant decrease in cancer cell viability [56,77]. This phenomenon was not observed in parental MCF-7 cells, which are not drug-resistant. Considering the increasing TKI-resistance, similar experiments were conducted to determine the effects of dopamine co-administration with another TKI, axitinib [57,78,79]. Dopamine enhanced the axitinib anti-cancer potency in the colony formation assay. Furthermore, a combination of axitinib and dopamine showed an additive effect on cancer cell apoptosis. We can assume that dopamine exerts a synergistic effect with axitinib via D1R activation. Nevertheless, no experiments with a selective D1R agonist were conducted, which could confirm the involvement of D1R in the described effects.

Dopamine D1 receptors seem to play a significant role in breast cancer cell motility, invasion, and subsequent metastatic potential. Fenoldopam inhibits the invasion of breast cancer cell lines MDA-MB-231 and BT-20 up to 70% [52]. D1 agonist (A77636) in the wound healing scratch assay reduced the cellular motility of MDA-MB-231 cell culture in a dose-dependent manner, and this effect was ameliorated by DRD1 gene silencing [55]. Yang et al. [53] also observed that fenoldopam and l-SPD reduced the speed of migration of breast cancer cells MDA-MB-231 and 4T1 in a dose-dependent manner. The reduction of cell migration was probably mediated by a decrease in the activity of RhoA and Rac1 GTPases [80].

Several studies have revealed the importance of cancer cell homeostasis and its factors such as cell volume and ECM stiffness on proliferation [81], migration [82], apoptosis [83], self-renewal properties [84], and drug resistance [85]. The idea of targeting cell volume regulation as factors affecting migration or metastases and a potential therapeutic anti-cancer target was previously discussed [86]. In the study constructed to investigate the impact of substrate stiffness on breast cancer cell volume homeostasis, the treatment of the MCF-7 cell line with D1R inhibitor (SCH23390) caused a decrease in cell volume, abolishing to some extent the increase in cell volume triggered by growing substrate stiffness [87]. Further validation of D1R antagonist effects on the migratory potential in cancer cells is demanded since cancer cells’ migration abilities in vitro generally reflect the metastatic abilities in vivo.

#### 3.1.2. In Vivo Studies

Borcherding et al. [52] showed that low and high doses (app. 10 nM and 30 nM in serum) of D1 agonist fenoldopam could reduce the volume of tumors up to 85% of control in a mouse xenograft model with MDA-MB-231 and SUM149 breast cancer cell line. Treatment leads to a 4-fold increase in apoptosis and a 2-fold increase in necrosis. Necrosis might result from the D1R-associated suppression of angiogenic pathways in tumors, which may explain more effective growth reduction in vitro than in vivo. Tumor growth suppression was sustained for at least two more weeks after fenoldopam administration. Since fenoldopam’s anti-tumor effect can exceed beyond the time of treatment, additional research is required to decide whether continuous infusion or shorter intermittent infusion is a more favorable scheme.

Significantly, reaching more than 30%, suppression of tumors’ growth and weight was observed after 18 days of treatment with A77636 (D1 receptor agonist) in a xenograft mammary tumor model with a 4T1.2 cell line [55]. Moreover, the antineoplastic activity of D1 agonist A77636 was investigated in the mouse model of bone metastasis [88]. D1R agonist treatment improved long bone structure- higher stiffness, denser trabecular structure, and decreased metastasis-associated bone degradation. The evaluation of A77636 activity on the histological structure of tibia trabecular bone reveals that this agent provided better outcomes in osteoclast-related bone-resorbing parameters than bisphosphonates. The discussed tridirectional activity of the D1R agonist (tumor-suppressive, bone anti-resorbing, and bone pro-forming) presents dopamine D1 receptor as a possible therapeutic target for preventing metastatic bone loss in breast cancer and preferably other tumors associated with bone metastases, such as prostate or kidney cancer.

Fenoldopam and l-SPD acting via D1 receptors significantly reduce the number of lung metastases without affecting the primary tumor’s weight in the orthotopic allograft mouse model with 4T1 breast cancer cells [53]. Both agents decreased the levels of epithelial–mesenchymal transition (EMT) markers: MMP-2 (matrix metallopeptidase 2) or E-cadherin. The reduced expression of E-cadherin is associated with the more mesenchymal-like phenotype of cancer cells. It can be followed by increasing invasive properties resulting in distant metastases from primary tumors [89,90]. The high expression of MMP-2 has also been described as an adverse prognostic factor for breast cancer metastases [91]. Fenoldopam decreased the neutrophils to lymphocytes ratio (NLR) in the peripheral blood of treated mice. Since neutrophils promote metastasis through assisting the formation of a pre-metastatic niche, its depletion can be acknowledged as a contributory mechanism of anti-metastatic D1R agonism activity, apart from the direct inhibition of the migratory potential of cancer cells [92,93]. Finally, lower NLR is in clinical practice associated with better survival among breast cancer patients [53].

Wang et al. [56] confirmed sunitinib and dopamine’s synergistic effect on tumors growth on a mice model bearing MCF-7/Adr xenografts. Dopamine exerted its effect via D1R. Most favorable tumor growth suppression is obtained when sunitinib and dopamine are administered to mice concurrently [94]. The investigation of the mechanistic background of in vivo dopamine action showed that this compound dose-dependently downregulated Wnt/β -catenin signaling downstream targets, cyclin D1, c-Myc, D-Myc, and activated apoptotic factors such as caspase 3, caspase 9 and Bax protein. Furthermore, dopamine has induced D1 receptor expression, which can be fundamental for the observed synergism of combined treatment. Supplementing the treatment with dopamine as an adjunct therapy leads to a significantly lower tumor burden compared with the monotherapy of axitinib (TKI) in the tumor mice model [57]. Combined treatment resulted in significantly smaller axillary lymph nodes, which seems to be an essential finding since lymphatic vessels are the most important way of spreading breast cancer [95]. Nevertheless, the histopathological analysis of lymph nodes was not conducted, so the observed enlargement might not result from metastases but rather cancer-associated inflammatory lymphadenopathy [96].

#### 3.1.3. Cancer Stem Cells

Among all dopamine receptors, D1R expression is upregulated in breast cancer cells’ population that displays a stem cell phenotype (CD44+ and CD24-) compared to the differentiated breast cancer cells [56,97]. A decrease in cancer stem cell (CSC) frequency was reported on the two metastatic breast cancer lines MDA-MB-231 and 4T1 as a result of treatment with fenoldopam and l-SPD, both in vivo and in vitro [53]. Furthermore, D1R agonist treatment resulted in a decrease in the expression of aldehyde dehydrogenase, which was considered an enzymatic hallmark of breast CSCs [98].

The addition of dopamine as an adjunct therapy to sunitinib treatment significantly reduced the number of breast CSCs in the MCF-7/Adr cell line in a dose-dependent manner via the induction of apoptosis [56]. This phenomenon was not obtained on the MCF-7 cell line, characterized by a low number of CSCs, which suggests direct anti-CSC action of dopamine. In breast cancer mouse models (MCF-7/Adr), the addition of dopamine to sunitinib abolished the induction of CSCs by sunitinib. Significant tumor growth suppression in combined (sunitinib + DA) treatment compared with sunitinib monotherapy comes partially from the preventive action of dopamine on the increase in CSC frequency that is triggered by sunitinib and many other drugs [53]. Furthermore, the addition of dopamine to axitinib resulted in a 3-fold decrease in the CSCs number in the tumor mouse model, the effect of which is probably mediated by the D1 receptor [57].

### 3.2. Nervous System Neoplasms

#### 3.2.1. In Vitro Studies

In 1990, Schrell et al. discovered that the D1 agonist SKF38393 significantly inhibited the growth of cell lines obtained from five different resected cerebral meningiomas at both 1 and 10 µM concentrations [60]. No further research, including in vitro experiments, on possibly targeting D1R to prevent the excessive growth of meningioma tumors was conducted.

DA reduces the viability of neuroblastoma cell culture SK-N-MC in the mechanism of dose-dependent, nitrile oxide (NO)-associated cell death induction, which is partially mediated by D1R agonism [99]. The effect is triggered to an approximately equal extent in two distinct pathways: 1) via D1 receptor activation and the subsequent enhancement of nitrile oxide (NO) production and 2) via the direct oxidant properties of DA. Nitrile oxide synthesis, in this case, could come from an increase in the activation of nitric oxide synthase (NOS) which in human occur in three distinct isoenzymes: inducible NOS (iNOS), endothelial NOS (eNOS) and neuronal NOS (nNOS) [100]. Enhanced production of NO upon treatment with D1 agonists resulted from overexpression of iNOS and, to a lesser extent, nNOS. Various kinases and pathways are involved in enhancing the NOS synthesis upon the activation of D1 receptors. These include cAMP and protein kinase A pathway, phosphatidylinositol 3-kinase/Akt cascade, and the NF-κB pathway. Interestingly, PKC’s role in the induction of NO was proven to be inhibitory, consistently with other reports [101]. Another study conducted on the same cell line showed that D1 receptor agonists (SKF38393 and DXH) activate p38 and c-Jun kinases and induce cAMP production in the PKA-dependent mechanism but do not cause the activation of ERK [102]. Even though the research mentioned above answered questions about the signaling pathway potentially triggered upon D1 receptor activation, it did not correlate and analyzed it in the context of neoplastic cell viability. Nevertheless, mechanisms should be investigated and described in further studies since contrary data are available regarding the ERK cascade in D1 receptor-mediated cytotoxicity on neuroblastoma cells [71].

In experiments conducted on the U87 glioma cell line, D1R agonist SKF38393 significantly promoted cancer cell proliferation and migration, while the D1R antagonist SCH23390 treatment had the opposite effect [103]. Another research provided contrary data where D1R agonist SKF83959 reduced the viability and migratory potential of U87 and U251 glioblastoma cell lines. These observations were confirmed on patient-derived cell lines. Presented data indicate that D1R activation increases intracellular Ca^2+^ and inhibits MAPK/mTOR signaling pathway. Altogether this impairs autophagic activity and leads to the accumulation of autolysosomes glioblastoma cells. Given the mortality and very limited therapeutic options for patients with glioma, further studies evaluating and extending available data should be considered, especially since one of the type 2 dopamine receptor modulators (ONC201) is already in clinical trials in malignant glioma [104].

#### 3.2.2. In Vivo Studies

D1R agonist SKF83959 has shown potency in the adjunct therapy of glioblastoma in the xenograft mice model [59]. A combination of temozolomide and SKF83959 resulted in the significant suppression of tumors growth compared with temozolomide alone. In vivo experiments confirmed results derived in vitro that D1R activation induced apoptosis via the impede of autophagic flux [59].

### 3.3. Lymphoproliferative Disorders

Stimulation of D1 receptors in activated T cells from healthy volunteers inhibits the proliferation of CD4+ and CD8+ lymphocytes in the cAMP-mediated mechanism [105,106]. The phenomenon mentioned above is absent in activated Jurkat T Leukemic Cells-D1 specific agonist fenoldopam in different concentrations (1 nM–6 µM) and did not decrease cell proliferation [107]. Performed experiments showed that it was caused by the high PDE activity, which in turn prevented intracellular cAMP accumulation, which may be the mechanism of resistance to D1 agonists anti-proliferative effects on leukemic T cells [108,109]. The introduction of potent PDE inhibitors combined with D1 agonists may help reduce leukemic T cells’ proliferation.

In the acute myeloid leukemia (AML), treatment with D1 agonist SKF38393 resulted in an over 90% decrease in cell number of two cell lines—AML-OCl2 and AML-OCl [110]. Yuan LB et al. [111] documented that D1 dopamine receptors are involved in the dopamine-induced apoptosis of K562 chronic myelogenous leukemia cells through the increase in intracellular cyclic AMP level [111].

### 3.4. Lung Cancer

#### 3.4.1. In Vitro Studies

Small cell lung carcinoma is considered as among the tumors in which development and progression may be modulated by the dopaminergic system [112]. It was shown on commercially available cell line H69 and cell cultures derived from patients’ tumors that treatment with fenoldopam does not affect the cell viability. However, this observation remains doubtable given the very short treatment time (3 h) and the fact that fenoldopam reduces bromodeoxyuridine incorporation in treated cells [113]. The investigation of this effect’s mechanism revealed that D1R activation led to a significant increase in cAMP and about a 2-fold increase in the level of DARPP-32 compared with basal level. It also caused a decrease in both Akt and p-Akt proteins. Interestingly, effects on DARPP-32 protein were even more evident when treatment was performed on the patient’s derived cell line that possesses dopamine D2 receptor gene polymorphisms with a rare allele in homozygosity (rs6275, rs6277), which brings the premise about the functional connection of D1 and D2 dopamine receptors [114].

#### 3.4.2. In Vivo Studies

Furthermore, dopamine treatment dose-independently decreased the frequency of non-small lung carcinoma CSCs in tumors from xenograft mice [115]. The authors hypothesized that dopamine’s effect results from D1 receptors’ activation even though no experiments confirming that theory were conducted. Dopamine and D1 receptors can be, after further research, considered as a part of biological machinery influencing lung cancer tumors, including non-small cell lung carcinoma.

### 3.5. Gastrointestinal System Cancers

#### 3.5.1. In Vitro Studies

The synergistic cytotoxic effect of N-arylpiperazine containing compound (C2) that possessed D1 agonist activity combined with sunitinib was reported in two cell lines of human pancreatic cancer SW1990 and PANC-1 [116]. In the colony formation assay, monotherapy with 2 µM C2 dose-dependently decreased the number of cell colonies (to 11.61% and 46.28% for SW1990 and PANC-1, respectively). D1R activation by C2 also resulted in a significant inhibitory effect on the migration of cancer cells. No other available D1R agonists were tested on pancreatic cell lines, which would be reasonable in the field of these findings. Moreover, it is unknown whether the activity of C2 involves other dopamine receptors besides D1R.

#### 3.5.2. In Vivo Studies

Su et al. [116] showed that the addition of N-arylpiperazine containing compound C2 significantly improved the sunitinib-induced inhibition of tumor growth in xenografts mice bearing SW1990 human pancreatic cell line tumors and patient-derived xenografts. Treatment with C2 or sunitinib in monotherapy did not show significant tumor growth inhibition, revealing the crucial role of the combination of these two drugs. The involvement of the activation of D1R by C2 was confirmed in an in vivo experiment where the intratumoral administration of D1 specific antagonist SCH23390 in the mice group treated with a combination of C2 and sunitinib resulted in the complete abolishing of growth inhibition. Moreover, both C2 and sunitinib can increase the expression of D1R, which may enhance the anti-proliferative effect of D1 receptor agonists. N-arylpiperazine containing compound showed effective anti-cancer properties and no evidence of systemic toxicity based on hemogram analysis, mice’s mass body changes, and morphological abnormalities in internal organs after treatment.

### 3.6. Other Tumors

The significance of D1R in the osteosarcoma role was established by Gao et al. [117]. They showed that D1R activation by SKF-38393 treatment and D1R protein overexpression significantly decreased the proliferation of the OS732 cell line by approximately 60%. Previous reports documented that D1R can play a crucial role in ERK1/2 and PI3K-Akt pathways activation, being the important modulator of cell proliferation [47,118,119]. In the osteosarcoma cells, the D1R activation decreased the phosphorylation of ERK1/2, PI3K, and Akt. Additional validation of the proposed mechanism revealed that the D1R activation effect on cell proliferation was similar to these achieved by treatment with ERK inhibitor (PD98059) and P3IK inhibitor (LY294002). Furthermore, the overexpression of D1R increased the rate of apoptosis, caspase-9 and -3 expression, enhanced the release of cytochrome c, and reduced the expression of anti-apoptotic protein Bcl-2 [120]. It also inhibited extracellular signal-regulated kinase 1/2 (ERK1/2) phosphorylation and induced the phosphorylation of p38 mitogen-activated protein kinase (p38 MAPK) and c-Jun N-terminal kinase (JNK).

## 4. D1 Receptor and Angiogenesis

The tumor’s angiogenesis blockage remains one of the essential directions in future anti-cancer therapies [121]. This pharmacological strategy is already being used to treat certain neoplasms, such as renal cell carcinoma [122]. Major studies showed that DA acting via D2R inhibits tumor angiogenesis by suppressing the actions of vascular permeability factor/vascular endothelial growth factor A on tumor endothelial cells and bone marrow-derived endothelial progenitor cells [123,124,125]. Consequently, the activation of D2R by dopamine suppressed stress-mediated tumor growth and the microvascular density of tumors while the activation of D1R did not influence these parameters in ovarian cancer stress mice models (SKOV3ip1 and HeyA8) [23,126]. A distinct role of the D1 receptor has been proposed. In mice bearing SKOV3ip1 ovarian tumors, dopamine, via D1R, causes microvessel maturation by increasing pericytes’ coverage. In vitro experiments confirmed that D1R explicitly mediates the stimulation of cell migration by dopamine on pericyte-like cell line (10T1/2) via the D1R-cAMP-PKA signaling pathway. Dopamine can be a pharmacological agent that can overcome and reverse favorable conditions for tumor angiogenesis that are a consequence of the increased concentration of norepinephrine and epinephrine in the plasma and tumor microenvironment triggered by chronic stress [127,128,129]. Not only anti-angiogenic but also agents normalizing the tumor vasculature were discussed as therapeutic targets in [130,131,132]. According to these findings, the treatment with dopamine and D1 receptor agonist SKF 82958 led to increased cisplatin tumor concentration due to normalizing the microvessel structure [126]. The conclusion is that D1R can be considered not only as a tumor cells-associated target but also as a microvasculature modulator (Figure 2).

## 5. Perspectives and Future Direction

Dopamine receptors’ ligands and their effects have been widely described and used in clinical practice, mainly in central nervous system disorders such as schizophrenia or Parkinson’s disease. Speculated effects of dopamine receptors on neoplastic cells brought us a new perspective on this neuropeptide system and novel insight into cancer biology, also in the context of purposing or even re-purposing dozens of dopaminergic ligands [133].

Multiple studies have shown the potential of dopamine receptors in the anti-cancer therapy of various tumor types using a non-selective activator of DRs, dopamine. The primary concern about the perspective utility of dopamine is its high concentrations used in most experiments that are possibly hard to achieve in vivo, while its physiological concentration is approximately 0.1 nM/l with daily fluctuations [134]. A high concentration of dopamine, used in many experiments, can involve other targets, not necessarily dopamine-specific receptors. Direct toxicity toward normal and cancer cells, resulting from, for instance, dopamine’s self-oxidative properties, should be excluded by the appropriate dosage of this agent in clinical trials. Furthermore, we are obliged to consider that most circulating dopamine in the human organism is sulfo-conjugated and thereby inactive [135]. Conversion to active form requires the presence of arylsulfatase A, in which concentration can vary depending on the organ and cancer type, making achieving the desired dopamine concentration even more difficult [136,137]. Another danger of using dopamine as an anti-cancer drug are its off-target side effects such as endocrine glands’ function impairment [138] or tachycardia [139].

The potency of activating D1R by its selective agonist was comprehensively analyzed in this review in the context of anti-cancer action. The anti-cancer properties of D1R can be understood as multidirectional since different beneficial mechanisms of its activity on cancer cells and tumors were described in multiple studies (Figure 2). At the level of single-cell molecular processes, D1R was involved in inhibiting cell viability by inhibiting the proliferation and induction of apoptosis. It also showed inhibitory potency on single-cell migration and presented its role in depleting CSC’s. Prospective insight into its roles on macroenvironmental tumors’ level allows us to indicate its role in the growth of xenografts tumors, distant metastases formation, tumors’ microvasculature maturation, or the increased distribution of cytotoxic drugs.

Furthermore, it is essential to bear in mind that the dopamine D1 receptor is involved in signaling pathways that differ depending on the tumor type. This phenomenon demands a critical approach toward studies on D1R, thus the existence of factors influencing the effectiveness of specific pathways, e.g., different PDE-4, -7, -8 activity among tumors in the cAMP-mediated pathway [140]. On the other hand, the effects of the modulation of D1 receptors can be enhanced by drugs targeting different components of its pathways, e.g., FDA-approved PDE5 inhibitors in the cGMP-mediated pathway in breast cancer [141]. The key role of tumors’ microenvironments and tissue-specific interaction must be emphasized as regulators of every cancer receptor-mediated signaling. For example, in adipose tissue D1R regulates the release of leptin, adiponectin or interleukin 6, which may contribute the development of the chemoresistance of this tumor [142,143]

In our review, we presented a wide range of dopamine D1 receptor agonists and antagonists varying in both selectivity and affinity to the targeted receptor and also pharmacological properties, such as bioavailability and half-time duration (Appendix A). Fenoldopam, a selective agonist of D1R, appears to be one of the most optimal candidates for further research. It is already in use as a hypotensive drug, as it possesses a high affinity to the receptor mentioned above (Kd = 2.3 nM), and does not cross the blood-brain barrier [144,145]. In the potential use of fenoldopam from a clinical perspective, the development of a slow-release formulation might be demanded due to its short half-time [146]. N-arylpiperazine containing compound (C2) is another D1R agonist with favorable oral bioavailability, making it utstanding among other modulators of D1R. Interestingly, N-arylpiperazine templates are usually applied to enhance the affinity of ligands in the development of agonists, which opens up new opportunities for the potential targeting of D1R [147,148]. Recently, another compound activating D1R, which is the D1/D5 agonist (PF-06649751), showed appreciable safety, tolerability, and a pharmacokinetic profile in clinical trials for Parkinson’s disease and could be successfully used in preclinical cancer research [149]. Since D1R is a target of some compounds already used in clinical practice (e.g., ecipopam or fenoldopam), it poses a possibility of screening drug libraries searching for D1R ligands. More important is research to invent new selective D1R agonist and antagonists, which may be used in the preclinical trials. Nevertheless, in every in vitro experiment investigating role of D1R, the comparison between continuous dosage and the intermittent short injection of the D1 agonist or antagonist should be carried out due to the possible desensitization and alteration of the expression pattern of these receptors that was already observed [150,151,152].

Insight into biological functions of D1R on cancer cells remains the most crucial issue in the perspective of the clinical application of pharmacologic targeting. Even though we could distinguish the effects of D1R agonism or antagonism in a wide range of tumors, many mechanisms responsible for it are still unsolved, both at the molecular and clinical levels. Studies referring to treatment with D1R agonists and antagonists lack the comparison group with normal epithelial cell lines adequate for certain cancers. It is crucial to exclude the toxicity of such agents towards noncancerous cells. Furthermore, no specified mechanism of overexpression of D1R in cancer tissues was proposed, such as proximity to an upregulated oncogene. Neither was it established what extent of protein overexpression was necessary for the satisfactory anti-cancer activity of the D1R agonist. To bring more in-depth insight into the role of this receptor overexpression, the use of cancer cell lines with induced D1R overexpression is essential in research. Hypothetically, the overexpression of dopamine D1 receptors can lead to apoptosis due to the activation of D1R by the physiological concentrations of dopamine. On the other hand, circulating dopamine can exert D1R agonism and subsequent anti-cancer properties in the neglectable range, due to too little expression of a targeted protein or other factors (e.g., activation of D2 receptors) that possess contradictory effects on cancer cells.

Future investigation should also be conducted in the context of D1R expression patterns in cancer metastases. Since it is known that D1R has a particular influence on migratory and invasive potential, the question is whether this receptor in metastases plays a different role compared with the primary tumor. It is also crucial whether D1R changes its impact on tumor functions during the initiation of carcinogenesis, further progression, and metastatic stages of neoplastic disease. Undoubtedly, the expression of D1R in metastatic tumors can vary between the primary site and metastases. Studies using distinct animal models dedicated specifically to metastasizing potency, e.g., the intracardiac injection of cancer cells, should be applied to better understand the role of D1R [153].

Even though the concept of cancer stem cells was proposed four decades ago, it is still unclear how to overcome the problem of self-renewing, as a small population of cancer cells is often resistant to traditional oncological drugs [154,155]. After a certain period, CSCs can reproduce cancer cells and lead to a recurrence of tumors, which remains a significant issue of clinical oncology, given the fact that many widely spread cytotoxic drugs, e.g., sunitinib, cyclophosphamide, or docetaxel, were shown to enrich the CSCs population in long-term treatment [156,157,158]. Nevertheless, targeting D1R receptors by its selective agonists brings the possibility of combined therapy with multitarget agents, cytotoxic drugs, or immunotherapy. Long-term advantages of such an approach might be beneficial, including reducing the drug dosage, minimizing side effects, or overcoming drug-resistance.

The dopaminergic system’s role in the context of tumor immunity and human immunology was noticed many years ago and is widely discussed [159,160]. Considering the D1 receptor’s role in cancerogenesis, it seems crucial to consider drug–tumor cell interaction and drug–circulating immune cells and the drug–tumor environment with infiltrating immune cells. Unfortunately, to our knowledge, all published research investigating the role of D1R in cancer biology focused only on the direct interaction of its ligands with cancer cells, neglecting the aforementioned possible connections. This phenomenon is a significant issue since there is contradictory evidence about the action of D1R on tumor immunity, where some indicate that these receptors’ activation boost the anti-cancer properties of immune cells, while the other points to the opposite effect. For instance, D1 receptors activation led to the inhibition of murine Gr-1 + CD115+ myeloid-derived suppressor cells (MDSCs) in a major interest in cancer research immunosuppressive properties such as suppressing effector T-cells’ anti-tumor activities and the expansion of tumor-specific pro-cancer regulatory T cells [161,162]. There are already several therapeutic approaches aiming for the depletion of MDSCs that could lead to the overcoming of tumor-induced immunosuppression in cancer [163]. On the contrary, nontoxic dopamine levels in lung carcinoma patients showed an inhibitory effect on CD4 and CD8+ T cells’ proliferation and cytotoxicity in a D1 receptor-mediated mechanism [164]. Such differences in the effects of D1R in anti-tumor immunity will be emerging with growing research on this topic, showing opposite conclusions depending on the immune cells’ population, the specificity of used dopamine receptor ligand, and its concentration. Thus, all in vivo experiments with dopamine receptors agonist/antagonist administration should be carefully analyzed since most animal models’ experiments were performed on immunodeficient mice. Nevertheless, observed tumor growth changes need to be considered as caused by direct D1R receptor modulation of cancer cells and blood vessels. Secondly, immune cells influence the development of the tumor. Furthermore, given the shared signaling pathways, the blockage or activation of D1R may significantly affect the immune system, including the already known side-effects of antipsychotic drugs (mainly D2 antagonists) such as agranulocytosis or pancytopenia [165,166]. A comprehensive approach in upcoming research, including evident cross-talk between D1R and the immune system, is required to establish receptor modulation’s fundamental pharmacological consequences. Understanding the role of the pharmacological administration of D1R modulators on immunity opens up new possibilities in combining natural anti-cancer treatment with widely spreading immunotherapy to achieve more satisfactory outcomes.

## 6. Conclusions

In summary, targeting D1R by its selective or non-selective agonist and antagonist was shown to have an anti-cancer activity that opens up new frontiers, especially when it comes to combined multi-targeted therapy. However, it has to be underlined that the effects of D1R activation can be tissue-specific. In light of available evidence, it is not possible to clearly establish its role in cancerogenesis. The discussed antineoplastic properties of D1R are suggestive but making it conclusive demands further research (Figure 3).

## Figures and Tables

**Figure 1 cancers-12-03232-f001:**
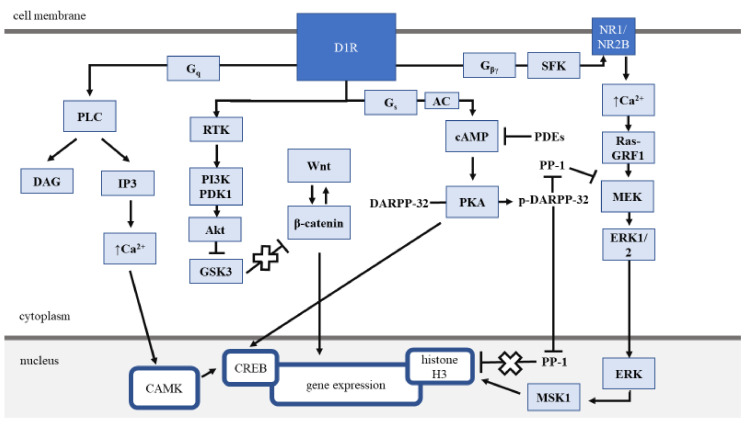
Dopamine D1 receptor signaling pathways. Dopamine D1 receptor exerts its function in both GPCR-mediated and GPCR-independent pathways. GPCR-mediated pathways include (1) the cAMP/PKA pathway, (2) the activation of phospholipase C, and (3) the ERK signaling pathway. Apart from the above, dopamine D1 receptor can activate Akt-GSK3 cascade via receptor tyrosine kinase. Nuclear responders of dopamine D1 receptor cellular pathways include, among others, the modulation of histone H3 and CREB activity. AC—adenylate cyclase, Akt—protein kinase B, CAMK—Ca^2+^/calmodulin-dependent protein kinase, cAMP—cyclic adenosine monophosphate, CREB—*cAMP-response element-binding* protein, D1R—dopamine D1 receptor, DA—diacylglycerol, DARPP-32—dopamine- and cAMP-regulated neuronal phosphoprotein, ERK1/2—extracellular signal-regulated kinase ½, G_q_, G_s_, G_βγ_—subunits of G protein, GRF1—guanine nucleotide-releasing factor 1, GSK3—glycogen synthase kinase 3, IP3—inositol triphosphate, MEK—mitogen-activated protein kinase, MSK—mitogen- and stress-activated kinase 1, NR1/NR2B—NMDA receptor subunits, PDEs—phosphodiesterase family, PDK1—phosphatidyl-dependent kinase 1, PI3K—phosphoinositide 3-kinase, PKA—protein kinase A, PLC—phospholipase C, PP-1—protein phosphatase-1, RTK—receptor tyrosine kinase, SFK—Src family kinase.

**Figure 2 cancers-12-03232-f002:**
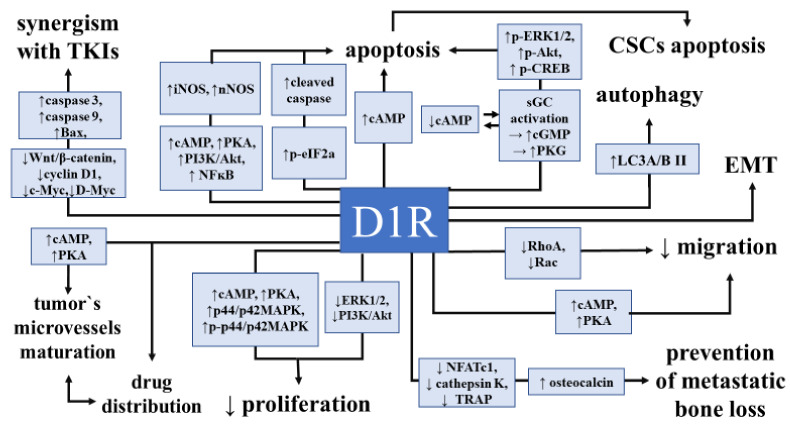
Role of dopamine D1 receptor in cancer biology and its signaling pathways. Modulation of the dopamine D1 receptor in neoplasm leads to various effects on cancer cells and tumors’ development and functioning. Cellular actions include the modulation of cell proliferation and apoptosis, autophagy induction, or a decrease in migration potency. Macro-scale effects are tumor’s vessel maturation, drug distribution in the tumor, and bone metastases formation. Akt—protein kinase B, Bax—Bcl-2-associated X protein, cAMP—cyclic adenosine monophosphate, cGMP—*cyclic* guanosine monophosphate, CREB—*cAMP-response element-binding* protein, ERK1/2—extracellular signal-regulated kinase 1/2, iNOS—inducible isoform of nitric oxide synthase, LC3A/B II—LC3-phosphatidylethanolamine conjugate, NFATc1—nuclear factor-activated T cells, cytoplasmic 1, NFĸB—nuclear factor kappa-light-chain-enhancer of activated B cells, nNOS—neuronal isoform of nitric oxide synthase, PI3K—phosphoinositide 3-kinase, PKA—phosphokinase A, PKG—protein *kinase G,* RhoA/Rac proteins—subgroups of the Ras superfamily of GTP hydrolases, sGC—*soluble* guanylyl cyclase, TRAP—tartrate-resistant acid phosphatase.

**Figure 3 cancers-12-03232-f003:**
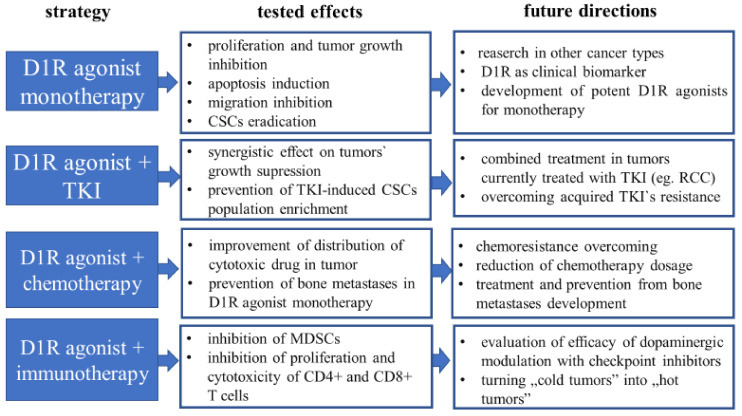
Future directions in research regarding targeting dopamine D1 receptor in cancer. Over the previous years, the knowledge about the dopamine D1 receptor role in neoplasm has been widening. Possible future directions in research and clinical implication in the translational medicine approach are presented. CSCs—cancer stem cells, D1R—dopamine D1 receptor, MDSCs—myeloid-derived suppressor cells, RCC—renal cell carcinoma, TKI—tyrosine kinase inhibitor.

**Table 1 cancers-12-03232-t001:** Summary of current data about the mRNA and protein expression of type 1 dopamine receptor and its clinical correlations for tumor subtypes. The name of the cell line or type of examined material is given in brackets.

Cancer Type	D1R Expression	Clinical Effects	Ref.
breast cancer	- protein (breast cancer patients’ samples, 4T1)—overexpression - protein (MDA-MB-231, MCF-7/Adr)—present - mRNA (MDA-MB-231,4T1, MCF-7/Adr)—overexpression	- D1R positive staining correlates with: - pre-menopausal age - estrogen receptor negative, progesterone receptor negative and HER2-possitive tumors - higher tumor stage - higher tumor grade - higher number of nodal metastases - shorter overall and recurrence-free survival	[52,53,54,55,56,57]
glioblastoma	- mRNA (U251)—upregulation - protein (patients’ samples)—downregulation	- patients whose tumors exhibit lower expression of D1R exhibit shorter median survival times	[58,59]
meningioma	- protein (patients’ derived cell lines)—present - mRNA (patients’ tissue samples, patients’ derived cell lines)	- expression showed in different histopathological types: meningotheliomatous, fibromatous, transitional and angiomatous meningioma	[60,61]
non-small cell lung carcinoma	NA	- DRD1 polymorphism (G allele rs686) predisposes to lung cancer among those exposed to secondhand smoke during childhood	[19]
seminoma	− mRNA (patients’ samples)—upregulation	- the D1 receptor expression, along with FAM71F2 (family with sequence similarity 71, member F2) expression, in the combined model, were proven to predict metastasized seminoma with a concordance of 87%, which can help to distinguish patients that do not require adjuvant therapy after surgery	[62]

D1R—dopamine receptor type 1, HER2—human epidermal growth factor receptor 2, NA—not available.

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
