# Peer review of "Dopamine D1 Receptor in Cancer"

_cancers, 2020, doi:10.3390/cancers12113232_

Round 1
Reviewer 1 Report
In their manuscript entitled ''Dopamine D1 receptor in cancer - a potential therapeutic target'', Sobczuk et al. summarize the current knowledge of the functions and the therapeutic targeting of dopamine D1 receptor in different cancers. The authors have done an excellent work, they have updated the current literature and have provided many examples from various cancer types. I have only some minor comments for corrections, which are summarized below :
- What about ongoing clinical trials using dopamine receptor agonists or antagonists? Maybe the authors can mention if there are already clinical trials in a small section in perspectives and future directions
line 46: cancers instead of cancer
line 53: in vivo, in vitro with italics (also there are more to be corrected, in the text)
line 84: delete ''this compound'' and replace it with DARPP-32
line 96: space between GSK3 and pathway
- abbreviations should be written with smaller letter and placed next to figure legends (for figures 1,2
line 143: delete ''all''
line 176: ''that'' instead of ''what''
line 188: ''et al'' instead of ''at all''
line 209: "which" instead of "what"
line 234: increased or decreased cell motility?
line 250: "in vivo" with italics
line 283: delete "level"
line 291: typo: through
line 328 and 329 "approximately" instead of "app."
line 339: as described
line 667: tumor growth changes
The authors should check more carefully for additional typos throughout the text.
Author Response
We want to thank the Reviewer for all efforts to review the manuscript. Below we answer all comments addressed to our work.
In their manuscript entitled ''Dopamine D1 receptor in cancer - a potential therapeutic target'', Sobczuk et al. summarize the current knowledge of the functions and the therapeutic targeting of dopamine D1 receptor in different cancers. The authors have done an excellent work, they have updated the current literature and have provided many examples from various cancer types. I have only some minor comments for corrections, which are summarized below :
We want to thank for the courteous comment.
What about ongoing clinical trials using dopamine receptor agonists or antagonists? Maybe the authors can mention if there are already clinical trials in a small section in perspectives and future directions?
To our best knowledge, the only compound among dopamine receptor agonists/antagonists that is currently in clinical trials is ONC201, a selective antagonist of D2 receptor. This fact is mentioned in the text (lines 297-299). There are no clinical trials with drugs targeting D1 receptors.
line 46: cancers instead of cancer
Revised as suggested.
line 53: in vivo, in vitro with italics (also there are more to be corrected, in the text)
Revised as suggested.
line 84: delete ''this compound'' and replace it with DARPP-32
Revised as suggested.
line 96: space between GSK3 and pathway abbreviations should be written with smaller letter and placed next to figure legends (for figures 1,2)
Revised as suggested.
line 143: delete ''all''
Revised as suggested.
line 176: ''that'' instead of ''what''
Revised as suggested.
line 188: ''et al'' instead of ''at all''
Revised as suggested.
line 209: "which" instead of "what"
Revised as suggested.
line 234: increased or decreased cell motility?
Revised as suggested.
line 250: "in vivo" with italics
Revised as suggested.
line 283: delete "level"
Revised as suggested.
line 291: typo: through
Revised as suggested.
line 328 and 329 "approximately" instead of "app."
Revised as suggested.
line 339: as described
Revised as suggested.
line 667: tumor growth changes
Revised as suggested.
The authors should check more carefully for additional typos throughout the text.
Spelling and grammar have been checked and corrected.
Reviewer 2 Report
The Review article entitled “Dopamine D1 receptor in cancer - a potential therapeutic target” by Sobczuk et al is a very straight forward review which provides lots of information regarding the role of D1 receptors in tumor growth and progression. However, there are certain areas that need improvement before the review can be published.
Language needs to be improved. Some sentences are too long and also grammatically incorrect, which disrupts the flow of reading.
Table 1: ‘Summary of current data about mRNA and protein expression of D1DR and its clinical correlations for tumor subtypes’; the tumor types where the clinical effects of D1 receptor expression is not known or determined should be excluded from the table.
‘Effects of D1 receptor in various tumor types’; this part of the review should be rewritten in a more concise manner. Similar findings can be clubbed together and under each tumor type, information can be sub divided into in vitro and in vivo results instead of subheadings like Cell viability, Migration and invasion….
Section 3.1.1 Breast cancer cell viability; the authors have discussed the effect of D1 receptor agonist SKF38393 on MCF-7 cells where they have mentioned the inhibitory effect of D1 agonist on MCF 7 cells ( Ref 57). However, in the next paragraph authors have mentioned that MCF7 cells were not susceptible to D1R agonist treatment which the authors found is consistent with previous observations (Ref 59). Please clarify.
Information has not been correctly interpreted at times. The effect of D1R on cell types and signaling cascades is context dependent. This should be taken into account while discussing the role of these receptors in different diseases.
Same paper has been discussed multiple times under different headings making the review unnecessary lengthy. Redundant information should be removed.
Author Response
We want to thank the Reviewer for all efforts to review the manuscript. Below we answer all comments addressed to our work.
The Review article entitled “Dopamine D1 receptor in cancer - a potential therapeutic target” by Sobczuk et al is a very straight forward review which provides lots of information regarding the role of D1 receptors in tumor growth and progression. However, there are certain areas that need improvement before the review can be published.
Language needs to be improved. Some sentences are too long and also grammatically incorrect, which disrupts the flow of reading.
The language has been checked and corrected to improve flow of reading.
Table 1: ‘Summary of current data about mRNA and protein expression of D1DR and its clinical correlations for tumor subtypes’; the tumor types where the clinical effects of D1 receptor expression is not known or determined should be excluded from the table.
We agree with a reviewer that the table should be shortened. We decided to minimalize Table 1 and exclude tumor types with no known clinical effects of D1 receptors. Nevertheless, information about expression of dopamine D1 receptor in different cancer types seems to be an important issue for this review thus we added Table S1. in supplementary materials with the data covering remaining tumor types in which no clinical correlations with D1 receptors have been established.
‘Effects of D1 receptor in various tumor types’; this part of the review should be rewritten in a more concise manner. Similar findings can be clubbed together and under each tumor type, information can be sub divided into in vitro and in vivo results instead of subheadings like Cell viability, Migration and invasion….
We agree with the reviewer, and as it was suggested, we substantially changed this part of the review. Headings,,in vivo studies” and ,,in vitro studies” were added and described accordingly. Similar findings were described together. This part has been shortened for the clarity of manuscript (lines 138-374) resulting in overall shortening of the manuscript from nearly 700 to 550 lines.
Section 3.1.1 Breast cancer cell viability; the authors have discussed the effect of D1 receptor agonist SKF38393 on MCF-7 cells where they have mentioned the inhibitory effect of D1 agonist on MCF 7 cells ( Ref 57). However, in the next paragraph authors have mentioned that MCF7 cells were not susceptible to D1R agonist treatment which the authors found is consistent with previous observations (Ref 59). Please clarify.
We agree that in the current form the information has been misleading. Treatment with D1 receptor agonist has no effect on MCF7 cell line when the growth is not induced by estradiol however it does when cells are pre-treated with estradiol. This issue was clarified in the corrected version of manuscript (lines 140-152)
Information has not been correctly interpreted at times. The effect of D1R on cell types and signaling cascades is context dependent. This should be taken into account while discussing the role of these receptors in different diseases.
We agree that the effect of D1R depends on many factors that need to be taken into account. In paragraph entitled ,,Conclusions” we emphasized that signaling cascades that dopamine D1 receptor is involved are dependent on the tumor type (534-539). The tissue specific action of D1R has been also underlined in lines 444-447. We believe that our discussion gives the reader impression that recent knowledge on role of D1R in cancer is not sufficient, but still growing, and should be analyzed with proper caution. Furthermore, we provide our concerns about uncertain aspects of this issue that could be resolved in the future if appropriate studies will be conducted. Our review may contribute to the understanding of the topic and be useful in the context for further researchers.
Same paper has been discussed multiple times under different headings making the review unnecessary lengthy. Redundant information should be removed.
Same paper have been discussed multiple times due to different aspects covered in the articles. Current manuscript is divided based on the tumor types and further into in vitro and in vivo studies. Thus some papers that covered both in vitro and in vivo part are cited in at least two different points. It is not possible no to discuss some papers multiple times but we have removed redundant information and changed the plan of the review with new headings. Due to extensive language editing and removal of redundant information (the length of article has been decreased by over 100 lines), most of these changes have not been marked in a track-changes mode.
Reviewer 3 Report
This manuscript purports to be a review article on on alterations in dopamine D1 receptor activity in known cancers and the potential of this receptor to serve as a target for therapeutic intervention in the treatment of cancer. From this reviewer's perspective, there are several issues that detract from the main strengths of the paper and which could be improved. They are addressed in order of approximate weight below:
1) The vast majority of the cited work appears to be derived from cell culture models of cancer. While this is problematic for any work attempting to extrapolate to in vitro possibilities, this is especially problematic for a receptor involved in cell-cell signalling to other tissues. It is virtually impossible to make any substantive conclusion regarding up or down-regulation of this receptor in cell culture because of this in the absence of context. As noted by the authors at line 650-651, none of the cited works extensively reviewed studied interactions with any tissue apart from the cancer or cancerous cell-line itself. Given the huge number of pathways that dopamine and dopamine D1 receptors affect (as noted by the authors themselves), this makes the data effectively meaningless.
2) The association of most of the cancers mentioned in the manuscript with alterations in D1 receptor protein are weak at best, making the blanket pretext for the work questionable in the first place. As the authors point out, the two cancers in which D1 receptor activity is most reliably known to be altered have not been studied for effects of D1 alteration. By far the best example seems to involve the use of the reasonably selective D1 agonist fenoldopam, which appears to reduce angiogenesis and subsequent cancer volume in animal models of human breast cancer. The title of the work suggests that the authors want to convince the reader that this is a good primary target for anti-cancer intervention. I for one am far from convinced of this idea.
3) For a strategy to work involving selective regulation of D1R expression or activity, the agonist needs to be selective, as the authors do note. However, not much attention is paid to any lead compounds toward that goal. Some mention is made of immuno-therapeutics, which to this reviewer are likely the best chance at reaching specificity given the state of the drug industry today and the number of small molecule inhibitors that have already been tested and rejected for dopamine receptors. Whether by modifying the fenoldopam backbone or using antibody techniques, this would seem to be a much higher concern than is given by the authors.
4) Numerous instances of grammatical and/or spelling errors detract from the message and are off-putting to making a sound argument. Line 588 (emphasis mine) "Fenoldopam, a selective agonist of D1R, appeals to be one of the most optimal candidate for research" agrees with what I take away from the literature, but should say "Fenoldopam, a selective agonist of D1R, appears to be one of the most optimal candidates for research". Likewise at 650 "...to our knowledge all researches investigating the role..." could and should be rewritten for clarity and proper word usage such as "...to our knowledge, all published research investigating the role..." or something similar. An established scientific reader of English text can interpret these without trouble, but they are apt to drive many bananas. The manuscript should be thoroughly proofread by a native English speaker prior to any re-submission.
The manuscript does have its strengths but I fear they are buried in an over-ambitious title and subsequent let-down to the keen reader. In summary, I feel the conclusions could be much more persuasive if presented even just from a slightly different perspective of adjunct therapy, as is highlighted toward the end of the paper. As a primary anti-cancer target, it seems rife with problems and insufficient data, data which is doubtful to alleviate many of said problems in the first place given the complex role of dopamine signalling pathways even when and if we do get it.
Author Response
We want to thank the Reviewer for all efforts to review the manuscript. Below we answer all comments addressed to our work.
This manuscript purports to be a review article on on alterations in dopamine D1 receptor activity in known cancers and the potential of this receptor to serve as a target for therapeutic intervention in the treatment of cancer. From this reviewer's perspective, there are several issues that detract from the main strengths of the paper and which could be improved. They are addressed in order of approximate weight below:
1) The vast majority of the cited work appears to be derived from cell culture models of cancer. While this is problematic for any work attempting to extrapolate to in vitro possibilities, this is especially problematic for a receptor involved in cell-cell signalling to other tissues. It is virtually impossible to make any substantive conclusion regarding up or down-regulation of this receptor in cell culture because of this in the absence of context. As noted by the authors at line 650-651, none of the cited works extensively reviewed studied interactions with any tissue apart from the cancer or cancerous cell-line itself. Given the huge number of pathways that dopamine and dopamine D1 receptors affect (as noted by the authors themselves), this makes the data effectively meaningless.
Our main goal of this review was summarizing recent knowledge on the role of dopamine D1 receptor in cancer and provide direction for further research rather than clinical implication of papers discussing dopamine D1 receptor in cancer. We agree that at the current stage it is almost impossible to find a clinical application for targeting D1 receptor in cancer, however available data shows that this receptor might be involved in the carcinogenesis. Thus, data presented in this review can be used as hypotheses-generating for future and current ongoing research. We underline these points in the discussion and conclusions (lines 534-539).
We believe that we used inappropriate title for our review suggesting clinical utility of D1 receptors for anticancer treatment which was misleading. For the clarity, the title has been corrected.
2) The association of most of the cancers mentioned in the manuscript with alterations in D1 receptor protein are weak at best, making the blanket pretext for the work questionable in the first place. As the authors point out, the two cancers in which D1 receptor activity is most reliably known to be altered have not been studied for effects of D1 alteration. By far the best example seems to involve the use of the reasonably selective D1 agonist fenoldopam, which appears to reduce angiogenesis and subsequent cancer volume in animal models of human breast cancer. The title of the work suggests that the authors want to convince the reader that this is a good primary target for anti-cancer intervention. I for one am far from convinced of this idea.
We agree that we used inappropriate title for our review suggesting clinical utility of D1 receptors for anticancer treatment which was misleading. For the clarity, the title has been corrected.
3) For a strategy to work involving selective regulation of D1R expression or activity, the agonist needs to be selective, as the authors do note. However, not much attention is paid to any lead compounds toward that goal. Some mention is made of immuno-therapeutics, which to this reviewer are likely the best chance at reaching specificity given the state of the drug industry today and the number of small molecule inhibitors that have already been tested and rejected for dopamine receptors. Whether by modifying the fenoldopam backbone or using antibody techniques, this would seem to be a much higher concern than is given by the authors.
We agree with the reviewer. We have decided to add additional supplementary table to present available agonists and antagonists of D1 receptors that could be used in future preclinical studies. We have stressed the superiority of fenoldopam (lines 450-455) and the need for further development of new agonists that are more selective (461-464). In the same paragraph we also gave examples of dopamine receptor agonist (ONC201 and PF-06649751) that are in clinical trials and mentioned possibility of drug screening in search for D1R ligands.
4) Numerous instances of grammatical and/or spelling errors detract from the message and are off-putting to making a sound argument. Line 588 (emphasis mine) "Fenoldopam, a selective agonist of D1R, appeals to be one of the most optimal candidate for research" agrees with what I take away from the literature, but should say "Fenoldopam, a selective agonist of D1R, appearsto be one of the most optimal candidates for research". Likewise at 650 "...to our knowledge all researches investigating the role..." could and should be rewritten for clarity and proper word usage such as "...to our knowledge, all published research investigating the role..." or something similar. An established scientific reader of English text can interpret these without trouble, but they are apt to drive many bananas. The manuscript should be thoroughly proofread by a native English speaker prior to any re-submission.
The language has been checked and corrected to improve flow of reading. Due to extensive language editing, this changes have not been marked in a track-changes mode.
The manuscript does have its strengths but I fear they are buried in an over-ambitious title and subsequent let-down to the keen reader. In summary, I feel the conclusions could be much more persuasive if presented even just from a slightly different perspective of adjunct therapy, as is highlighted toward the end of the paper. As a primary anti-cancer target, it seems rife with problems and insufficient data, data which is doubtful to alleviate many of said problems in the first place given the complex role of dopamine signaling pathways even when and if we do get it.
We agree with this comment. There is not enough data to suggest clinical applicability of D1 targeting in cancer treatment. We have changed the title accordingly, as well as abstract and conclusions to underline possible involvement of D1 receptors in cancer development and progression and hypothetical use of D1 targeting as adjunct therapy, rather than primary anti-cancer treatment.
Round 2
Reviewer 3 Report
I think it is much stronger now and not misleading in scope.